Modeling geographic distribution of arbuscular mycorrhizal fungi from molecular evidence in soils of Argentinean Puna using a maximum entropy approach

Nepote Valentin Davide 1
Voyron Samuele 1 2
http://orcid.org/0000-0003-2189-8529 Soteras Florencia 3
Iriarte Hebe Jorgelina 4 5
Giovannini Andrea 1
http://orcid.org/0000-0001-5864-8561 Lumini Erica 2 erica.lumini@ipsp.cnr.it
Lugo Mónica A. 4 5
1 Life Sciences and Systems Biology, University of Turin , Torino , Italy
2 Institute for Sustainable Plant Protection (IPSP), National Research Council (CNR) , Torino , Italy
3 Laboratorio de Ecología Evolutiva y Biología Floral, Instituto Multidisciplinario de Biología Vegetal (IMBIV), CONICET, FCEFyN, Universidad Nacional de Córdoba , Córdoba , Argentina
4 Instituto Multidisciplinario de Investigaciones Biológicas (IMIBIO-CONICET-UNSL) , San Luis , Argentina
5 Micología, Diversidad e Interacciones Fúngicas (MICODIF), Área Ecología, Facultad de Química, Bioquímica y Farmacia, Universidad Nacional de San Luis (UNSL) , San Luis , Argentina
Góes-Neto Aristóteles
Electronic publication date: 2023 Jan 12
Publication date: 2023
Volume: 11
Electronic Location ID: e14651
Received 2022 Jun 8; Accepted 2022 Dec 7
Copyright: © 2023 Nepote Valentin et al.
Copyright year: 2023
Copyright holder: Nepote Valentin et al.
License: This is an open access article distributed under the terms of the Creative Commons Attribution License, which permits unrestricted use, distribution, reproduction and adaptation in any medium and for any purpose provided that it is properly attributed. For attribution, the original author(s), title, publication source (PeerJ) and either DOI or URL of the article must be cited.
License URL: https://creativecommons.org/licenses/by/4.0/

Keywords: Argentinean Puna, Species distribution modeling (SDM), Fungal metabarcoding, Arbuscular mycorrhizal fungi (AMF), MaxEnt

Funding: PROICO 2-2718 Consiglio Nazionale delle Ricerche (CNR)/Consejo Nacional de Investigaciones Cientıficas y Tecnologicas (CONICET) PIP 833 (CONICET) This work was supported by PROICO 2-2718 (Facultad de Quımica, Bioquımica y Farmacia, Universidad Nacional de San Luis), by Italia/Argentina Bilateral Joint Research Project Consiglio Nazionale delle Ricerche (CNR)/Consejo Nacional de Investigaciones Cientıficas y Tecnologicas (CONICET) ‘Ecological characterization of Arbuscular Mycorrhizal Fungi communities as ecosystem indicators for arid and semiarid Argentinean soils’ (2017–2019) and PIP 833 (CONICET). The funders had no role in study design, data collection and analysis, decision to publish, or preparation of the manuscript.

==============================
The biogeographic region of Argentinean Puna mainly extends at elevations higher than 3,000 m within the Andean Plateau and hosts diverse ecological communities highly adapted to extreme aridity and low temperatures. Soils of Puna are typically poorly evolved and geomorphology is shaped by drainage networks, resulting in highly vegetated endorheic basins and hypersaline basins known as salar or salt flats. Local communities rely on soil fertility for agricultural practices and on pastures for livestock rearing. From this perspective, investigating the scarcely explored microbiological diversity of these soils as indicators of ecosystems functioning might help to predict the fragility of these harsh environments. In this study we collected soil samples from 28 points, following a nested design within three different macro-habitats, i.e., Puna grassland, hypersaline salar and family-run crop fields. Total fungi and arbuscular mycorrhizal fungi (AMF) occurrence were analyzed using eDNA sequencing. In addition, the significance of soil salinity and organic matter content as significant predictors of AMF occurrence, was assessed through Generalized Linear Mixed Modeling. We also investigated whether intensive grazing by cattle and lama in Puna grasslands may reduce the presence of AMF in these highly disturbed soils, driving or not major ecological changes, but no consistent results were found, suggesting that more specific experiments and further investigations may address the question more specifically. Finally, to predict the suitability for AMF in the different macro-habitats, Species Distribution Modeling (SDM) was performed within an environmental coherent area comprising both the phytogeographic regions of Puna and Altoandino. We modeled AMF distribution with a maximum entropy approach, including bioclimatic and edaphic predictors and obtaining maps of environmental suitability for AMF within the predicted areas. To assess the impact of farming on AMF occurrence, we set a new series of models excluding the cultivated Chaupi Rodeo samples. Overall, SDM predicted a lower suitability for AMF in hypersaline salar areas, while grassland habitats and a wider temperature seasonality range appear to be factors significantly related to AMF enrichment, suggesting a main role of seasonal dynamics in shaping AMF communities. The highest abundance of AMF was observed in Vicia faba crop fields, while potato fields yielded a very low AMF occurrence. The models excluding the cultivated Chaupi Rodeo samples highlighted that if these cultivated areas had theoretically remained unmanaged habitats of Puna and Altoandino, then large-scale soil features and local bioclimatic constraints would likely support a lower suitability for AMF. Using SDM we evidenced the influence of bioclimatic, edaphic and anthropic predictors in shaping AMF occurrence and highlighted the relevance of considering human activities to accurately predict AMF distribution.

Introduction

Overview of the ecosystems of Puna

The South American biogeographic region of Puna extends within the Andes Mountain Range across Argentina, Bolivia, Perú and Chile at elevations higher than 3,000 m. In the Quechua local native language Puna means “high and cold lands” (Carilla, Grau & Cuello, 2018). This meaning well matches the broad concept of Puna, which includes biogeographical, ecological and geographical standpoints. These high Andean plateau regions encompass diverse ecological communities which share severe or extreme aridity and wide seasonal and daily temperature variations (Lugo & Menoyo, 2019 and references therein). Within Argentinean boundaries, the Puna geological province (Turner & Méndez, 1979) is the geomorphological basement encompassing different but largely overlapping ecological classifications i.e., Puna region (Morrone, 2001), Puna and Altos Andes ecoregions (Matteucci, 2018a, 2018b), phytogeographic provinces of the Dry, Humid and Desert Puna (Troll, 1959, 1968), phytogeographic provinces Puneña and Altoandina (Cabrera, 1976; Cabrera & Willink, 1980; Carilla, Grau & Cuello, 2018; Oyarzabal et al., 2018), and Puna floristic districts “Jujeño”, “Central”, “Cuyano”, and “Boliviano” (Martínez Carretero, 1995). Since the above mentioned phytogeographic provinces and ecoregions are extending along similar territories, hereafter these will be named Puna and Altoandino. Within Altoandino, a location occupied by local farmers is included in this research, namely Chaupi Rodeo (Jujuy, Argentina). Puna and Altoandino are distinguished mainly by their specific elevations, reaching 3,000–3,400 m and 4,000–4,500 m, respectively (Carilla, Grau & Cuello, 2018; Cabrera, 1976; Oyarzabal et al., 2018). Despite the differential altitudinal range and its relative effect on vegetation features, similar geological, historical and ecological processes (Matteucci, 2018a, 2018b) led some authors to consider Puna and Altoandino as a unique and homogeneous macro-area. As a whole, common environmental conditions such as low temperature climate, severe aridity and wide daily and seasonal temperature range make this area a quite uniform region where the study of the distribution patterns of fungal communities in relation to abiotic variables and different habitats can be insightful for soil mycobiota research.

Puna and Altoandino host shrub-dominated vegetation units and gramineous steppe, as well as other azonal communities such as halophyte vegetation within hypersaline endorheic basins named Salinas or salar and plant communities dominated by Poaceae, Juncaceae, Cyperaceae within wet flooded endorheic basins called Vegas (Martínez Carretero, 1995; Renison et al., 2013; Carilla, Grau & Cuello, 2018; Matteucci, 2018a, 2018b; Oyarzabal et al., 2018).

The human impact on Puna soils

Soils of Puna and Altoandino are typically poorly evolved, influenced by cryogenic processes and by aridity (Panigatti, 2010). These ecosystems are highly exposed to desiccation, extreme environmental conditions such as large daily temperature amplitudes, an incident solar energy greater than 2,200 KW/m2/year and intense UV irradiation (Martínez Carretero, 1995; Carilla, Grau & Cuello, 2018; Matteucci, 2018a, 2018b). These harsh conditions result in high community vulnerability when exposed to sudden ecological changes, mainly caused by the anthropic impact of extensive grazing of camelids and cattle, wildfires, mining and wood harvesting (Carilla, Grau & Cuello, 2018; Matteucci, 2018a, 2018b). Consequently, increasing desertification processes occur at an extremely high level within these ecosystems (Vorano & Vargas Gil, 2002). With regard to extensive grazing, livestock management in these areas usually follows the nomadic type with continuous migration of multispecies herds composed by sheeps, goats and domesticated llamas and alpacas (Vorano & Vargas Gil, 2002; Quiroga Mendiola & Cladera, 2018), which add up their impact to the grazing of wild species of camelids (huanaco, Lama guanicoe Statius Müller and vicuña, Vicugna vicugna Molina). Few areas are impacted by cattle and camelid flocks circulation, restricted by paddocks or corrals (Carilla, Grau & Cuello, 2018). This migratory grazing system likely turned progressively natural grasslands into shrublands. Environmental exploitation impacts on soil fertility and ecosystem services that local subsistence-based native communities rely on, with direct effects on their economy and living conditions. In this perspective, investigating the microbiological diversity of these soils might be a valuable indicator of the ecological conditions and of the ecosystem’s disturbance, therefore helping to evaluate the fragility of these natural ecosystems.

Species distribution modeling of arbuscular mycorrhizal fungi in argentinean Puna

Soil-borne fungi play a fundamental role in the ecosystem functioning, as decomposers of organic matter, inducing soil aggregation, as pathogens, or mutualists enhancing plant growth. Among soil fungal communities and functional guilds, arbuscular mycorrhizal fungi (AMF) are worldwide obligate plant symbionts belonging to Phylum Glomeromycota (Wijayawardene et al., 2020) promoting their growth by improving soil nutrients and water uptake as well as providing pathogen protection to their host (Smith & Read, 2008). Despite the AMF widely assessed ubiquity, their occurrence and abundance may be affected by environmental conditions (e.g., soil nutrients, pH, precipitation, temperature; Tedersoo et al., 2014; Kohout et al., 2015; Fitzsimons, Miller & Jastrow, 2008; Dumbrell et al., 2010) and vegetation type (Veresoglou, Caruso & Rillig, 2013; Davison et al., 2021). For instance, in a global analysis Tedersoo et al. (2014) evidenced a positive relationship of AMF richness with potential evapotranspiration and soil pH, as well as a diversity increase in grassland and shrubland ecosystems. Therefore, AMF richness and distribution are expected to be affected by current land uses, such as cropping or grazing, and the study of the AMF community patterns in relation to climatic and edaphic conditions may help to predict their response to climate change (Kivlin et al., 2017) scenarios. Species distribution modeling (SDM) combines occurrence data with environmental variables to geographically predict potential suitable areas within the studied environments (Elith & Graham, 2009). This approach has been rarely used for estimating AMF distribution (Kivlin et al., 2017), mainly because of the difficulty in delimiting the potential area of occurrence for these soil-borne organisms. However, considering that geological, environmental and soil conditions have delimited the geographical boundaries of our study area, the ecosystems of Puna include promising regions to predict AMF distribution.

Mycorrhizal fungi, and more specifically the AMF which are the focus of this research, are main drivers of soil nutrients and plant-soil ecology (Wurzburger et al., 2017). Soils of Puna bear particular chemical, physical and anthropic features and represent an ideal target for mycorrhizal symbionts comparative analysis. Considering the importance of AMF in plant-soil dynamics (Davison et al., 2021; Dumbrell et al., 2010; Bonfante & Genre, 2015), the paucity of researches on the fungal profile of Puna soils and the lack of predictive approaches on AMF spatial suitability, in the present study we focused on AMF, relying on the fungal OTU (Operational taxonomic Units) table provided in Ontivero et al. (2020) and extending the analysis to additional soil sampling points collected in grazed or undisturbed areas and in an endorheic salar basin, within non-cultivated Puna environments.

Research questions, analysis and modeling

The aim of our work was to reliably predict the suitability of the three studied macro-habitats of the Argentine Puna for AMF, by evaluating their relative abundance within the framework of the total soil fungal profile. For this reason, according to the results of Berruti et al. (2017), we preferred the use of fungal generic primers targeting the ITS2 region instead of specific AMF primers. The main idea is that the assessment of AMF potential distribution may represent a useful proxy of diverse ecological processes and plant-soil mycobiota interactions. This may also be a first step in understanding which of the current human interventions may be more impacting and detrimental to the Puna ecosystems, in a context where local native communities must maintain a frail balance between subsistence and conservation. To do so, we first estimated the correlation of the measured soil parameters and land uses with AMF occurrence in the different locations and habitats through a GLMM approach, then we performed a model selection on GLMM outputs, weighting and averaging multiple models by means of Multi Model Inference (MMI). Finally, we performed a SDM maximum entropy approach (Bradie & Leung, 2017; Zimmermann et al., 2010; Austin, 2002) in order to predict the environmental suitability for AMF in the Argentinean Puna and Altoandino, phytogeographically and ecoregionally delimited as mapped by Oyarzabal et al. (2018), and, according to Matteucci (2018a, 2018b) and Morello et al. (2018), we considered them as a unique geological province and bounded it above 27° latitude S to maintain a relatively homogeneous area of prediction. Since Chaupi Rodeo is located in a fringe of Altoandino extending into Puna and the current human agricultural use of soil in this area might have influenced AMF communities, we developed different models, with and without the Chaupi Rodeo sampling points, and compared the suitabilities of the predicted areas for AMF.

Materials and Methods

Soil sampling and environmental metadata

Soil samples have been collected from 28 sampling points along transects within six different locations in the northern Argentinean Jujuy Province belonging to three main different macro-habitats: (i) twelve sampling points were chosen in grassland/shrubland habitats of Puna (four sampling points for each of the following locations, named Dunas, Punto Susques, Puesto del Marqués, Abra Pampa), (ii) three sampling points in a hypersaline salar basin area (location Salinas Grandes), and (iii) nine sampling points within family-run crop fields in three Chaupi Rodeo locations in Altoandino (Ontivero et al., 2020), from now on named Chaupi Rodeo A, B and C (Fig. 1). Each Chaupi Rodeo location consisted of a Vicia faba field (named CRF), a potato field (named CRP) and a corn field (named CRM). Five soil subsamples per each sampling point were therefore collected from each of the nine crop fields in the three locations mentioned above. Multiple environmental explanatory variables have been considered per each sampling point, namely elevation, grazing type (i.e., undisturbed habitats or grazed by Lama and/or cattle), preceding habitat or crops in the previous year (as recorded during the cropfields survey and considering salar and Puna habitats as unchanged from the previous year) and plant cover (on a qualitative scale as assessed during the field survey). Soil physicochemical analyses were carried out in the Soil Laboratory of INTA (National Institute of Agricultural Technology), EEA (Experimental Agricultural Station) Villa Mercedes, San Luis, Argentina, following the standards established by the IRAM (Argentine Institute for Standardization and Certification) dependent on the Secretary of Agriculture, Livestock and Fisheries (SAGyP) of the Agency of Economy of Argentina, which provides the methodological protocols to be applied to the different soil types in the Country (for detailed description of protocols, see https://www.magyp.gob.ar/sitio/areas/samla/normas/). Each soil sample was analyzed for pH (ratio 1 in 2.5), soil carbon quantity (as organic carbon by the wet method of Walkley and Black, modified by Richter, estimating oxidation at 77%, and expressed in grams %), percentage of organic matter (Walkley Black method expressed in %, multiplied soil carbon quantity by 1.724), available phosphorus (Bray and Kurtz method), total nitrogen (per micro Kjeldahl expressed in mg.gr) and electrical conductivity measured as dS/m was used as a proxy for the salinity class of soils (such as not saline: 0–2 dS/m, slightly saline: 2–4 dS/m, moderately saline: 4–8 dS/m, very saline: 8–16 dS/m, and extremely saline: >16 dS/m). Precise geolocation for each sampling point was recorded. Sampling was authorized by: Gobierno de Jujuy, Ministerio de Ambiente, Jujuy Argentina RESOLUCION n 024/2018-S.B.

Figure 1 Geolocation of the analyzed sampling points within the Argentinean Jujuy province.

Geographical overview of the sampling points within the six sampling transect named as follows: Salinas Grandes, Dunas, Punto Susques, Abra Pampa, Puesto del Marqués, Chaupi Rodeo (which includes three cropfield locations labeled as A, B, C). The legend in the box below indicates the boundaries of Argentinean Puna and Altoandino. The sampling points color legend indicates the different macro-environments of salar, Puna grasslands and crop fields. The legend of Chaupi Rodeo sampling points indicates the different crop types cultivated for the three locations A, B, C. It is worth to be noted that floristic, biogeographical and ecological ecosystems of the Puna biome exceed the limits of the Puna geological province located at Catamarca Province (Turner & Méndez, 1979) and extend south to the Mendoza Province (Martínez Carretero, 1995; Matteucci, 2018b) in Argentina. Attribution for the Google Satellite images used in QGIS processing: Map data ©2015 Google.

Sequencing and bioinformatic analysis

The five soil subsamples taken from each sampling point were homogenized and separated into fractions for soil physicochemical and molecular analyses. DNA extraction, PCR amplification, amplicons sequencing and bioinformatic analysis have been performed as described in Ontivero et al. (2020). Soil samples were sieved through a 2-mm mesh size sieve. Genomic DNA was extracted in three replicates per soil sample, each replicate weighing 250 mg, by means of the DNeasy PowerSoil kit (Qiagen, CD Genomics Company, Shirley, NY, USA) according to the manufacturer’s instructions. To investigate both the AMF and the total fungal biodiversity, the ITS2 region was amplified by a nested PCR approach, which has been demonstrated providing enough data to estimate community structure, without major biases, and to make reliable conclusions on AMF relation to environmental variables (Berruti et al., 2017). Briefly, the nuclear ribosomal region was amplified with the generic fungal primer pair ITS1F, White et al. (1990) and ITS4, Gardes & Bruns (1993). The PCR products obtained were then processed for a semi-nested PCR approach targeting the ITS2 region, with primer fITS9 Ihrmark et al. (2012) and ITS4. All the primers employed in the nested PCRs were added of the Illumina overhang adapter sequences: forward overhang: 5′-TCGTCGGCAGCGTCAGATGTGTATAAGAGACAG- [locus specific target primer], reverse overhang: 5′ GTCTCGTGGGCTCGGAGATGTGTATAAGAGACAG- [locus specific target primer]. PCR products were checked on agarose 1% gel and purified using the Wizard SV Gel and PCR CleanUp System (Promega, Madison, WI, USA). Prior to Illumina MiSeq (2 × 300 bp), performed by BMR Genomics (Padova, Italy), samples were quantified using Qubit 2.0 (Thermo Fisher Scientific, Waltham, MA, USA). Forward and reverse reads were preprocessed eliminating Illumina adapters and primers sequences and then analyzed by means of the bioinformatics platform QIIME2 (Quantitative Insights Into Microbial Ecology 2) version 2019.7 (Bolyen et al., 2019). In order to retain high quality sequences and to remove chimeras the plugin DADA2 (Callahan et al., 2016) was adopted. More in detail, for quality control, sequences were truncated at 280 bp for forward and at 265 bp for reverse and chimeras were removed using the “pooled” methods. The plugin qiime vsearch cluster-features-de-novo, using 97% as the identity threshold, was used to obtain OTUs tables. The taxonomic assignation of retrieved fungal community was obtained by means of a classifier built using the last release of UNITE Community (2019): UNITE QIIME release for Fungi version 10.05.2021 (Abarenkov et al., 2021). The trophism of the fungal community was assessed by means of FUNGuild (Nguyen et al., 2016). The taxonomy assignment was also checked doing a BLAST analysis against the MaarjAM database (https://maarjam.ut.ee/; Öpik et al., 2010). The datasets originated (OTU table, taxonomy and metadata) were then used to build two phyloseq object, R package phyloseq v. 1.36.0 (McMurdie & Holmes, 2013). The Krona Tools and cpauvert/psadd library (Ondov, Bergman & Phillippy, 2011). Were used to generate Krona plots. The raw data of this study are deposited in the NCBI Sequence Read Archive (SRA-NCBI; https://www.ncbi.nlm.nih.gov/sra) under project accession number PRJNA835719.

Generalized linear mixed modeling and multivariate analyses

To analyze the AMF communities, the OTU table was rarefied at an even sequencing depth of 19,882 sequences per sample and restricted to the Phylum Glomeromycota. Due to a non-normal distribution of data, a non-parametric Kruskal-Wallis analysis of variance was performed to test for the difference among locations using the function kruskal.test in the R stats package version 4.1.0 (R Core Team, 2013). Post-hoc pairwise tests were then performed applying a Bonferroni correction by using the function pairwise.t.test in the R Stats package.

To assess the influence of bioclimatic variables and soil parameters (i.e., predictors) on AMF reads, and considering the nestedness of the sampling design and the overdispersion of data, a negative binomial GLMM (Generalized Linear Mixed Model) approach was then performed, using the function glmer.nb in the R package blmeco version 1.4. (Korner-Nievergelt et al., 2015), including samples nested within locations as a random term. Collinearity among predictors was previously tested by means of Variance Inflation Factor approach using the R package car version 3.0–12 (Fox & Weisberg, 2019). A Pearson’s r coefficient value of 0.7 was chosen as a collinearity threshold. The model selection was performed both through a Minimum Adequate Model and a Multi Model Inference approach using the R function drop1 and the R function dredge in the package MuMIn version 1.43.17 (Barton, 2020). The sum of weights for each predictor was estimated from the best competing models, then model-averaged coefficients were estimated and the difference from zero tested for significance. Beta-diversity among different locations and among habitats was assessed by means of Non-metric Multi-Dimensional Scaling (NMDS) ordination system using Jaccard dissimilarity index and statistical significance was verified performing a PERMANOVA test, using the software MycrobiomeAnalist (Chong et al., 2020; Dhariwal et al., 2017).

Geographical information systems and species distribution models

A Species Distribution Model approach was used to model the potential distribution of AMF by estimating the environmental suitability through the maximum entropy-based software MaxEnt (Phillips, Dudik & Schapire, 2004; Phillips, Anderson & Schapire, 2006; Elith et al., 2011; Warren & Seifert, 2011) version 3.4.1. The AMF suitability was modeled within the area delimited by Puna and Altoandino boundaries as previously described, following the mapping in Oyarzabal et al. (2018). Due to similar tectonic and orogenic dynamics, Argentinean Puna and Altoandino can be considered as a unique region northern of 27° latitude S (Morello et al., 2018), allowing to model the environmental suitability for both. Consequently, the estimated area was bounded southward by this latitude (Matteucci, 2018a, 2018b) to avoid excessive environmental heterogeneity for the predictions. Cartographic processing was performed through the software QGIS version 3.16 (QGIS Development Team, 2021) using the World Geodetic System WGS84 for rasters and vectors. A set of climatic and environmental rasters were used as predictors of the habitat suitability for AMF. Nineteen Worldclim 2 (Fick & Hijmans, 2017) bioclimatic variables, regarding temperature and precipitation, and elevation at maximum resolution of “30” were chosen as predictors along with the following environmental rasters: Land Cover at “30” resolution by National Mapping Organizations—GLCNMO based on index LCCS developed by FAO; vegetation units (Oyarzabal et al., 2018) corresponding to Puna (dominated by Fabiana densa Remy and Baccharis boliviensis (Wedd.)) Cabrera and Altoandino (dominated by Senecio algens Wedd. and Oxalys compacta Gillies ex Hook. & Arn.); soil nutrients retention and soil nutrients availability by FAO Geonetwork, and soil type by SMW-Digital Soil Map of the World—FAO (Fischer et al., 2008). The selection of climatic variables was performed stepwise and climatic variables collinearity was tested by means of the R package corrplot version 0.92 (Wei & Simko, 2021) at a threshold value of 0.7. Multivariate Principal Components Analysis (PCA) was performed on the values of the bioclimatic variables extracted per sampling point, using the R package factoextra (Kassambara & Mundt, 2020).

A first set of models were generated through MaxEnt using the 19 Worldclim bioclimatic variables. Permutation importance values (as a percent on the total importance) and jackknife AUC (area under the receiver operating curve) bar plots of the model performance, including or excluding each variable, were used for a stepwise selection of bioclimatic predictors by cross-referencing them with PCA vectors and the clusters of the plotted sampling points. The selected bioclimatic variables were added to the above described environmental and edaphic predictors and a new set of 10 bootstrap MaxEnt models was generated optimizing the process to maximum 5,000 iterations and 10,000 background points and setting the model convergence threshold to 0.0001. The model evaluation was performed choosing as test data the 30% of the sampling points through an iterative random seed bootstrap process for each model run (Nguyen et al., 2021; Reddy et al., 2015; Préau et al., 2018). A stepwise process of model selection was then performed by comparing AUC values, permutation importance of variables and jackknife AUC test results. Basing on Maxent generated threshold-dependent binomial tests of omission rates (Phillips, Anderson & Schapire, 2006), presence-absence thresholds with the lowest omission rates were chosen for the cumulative and the cloglog outputs, respectively as a mean of fixed cumulative 10% values of threshold, and as a mean of the thresholds generated by balancing training omission, predicted areas and threshold values. Since all the Chaupi Rodeo points of presence were sampled in agricultural soils, while the other samples were collected from unfarmed habitats of Puna and salar, a new model was generated with the same variables previously selected but excluding the Chaupi Rodeo points of presence. This last step not only accounts for the possible biases in predicting AMF suitability when considering both cultivated and uncultivated soils but also allows to investigate the differences between the predicted habitat suitability for AMF in the Chaupi-Rodeo area when using all points and when excluding the points collected in cultivated soils.

Results

Total fungal abundances and beta-diversity

The taxonomic analysis of the retrieved fungal communities showed that the highest number of reads belonged to the Phylum Ascomycota with more than 30,000 sequences per location, followed by Phylum Basidiomycota, ranging from more than 8,500 up to almost 32,000 sequences across samples (Fig. 2, Table S1). The Phylum Glomeromycota showed a high abundance per sample above 900 sequences only in the croplands of Chaupi Rodeo, with the exclusion of potato fields, and in the sampling point Abra_D in Abra Pampa (Fig. 2, Table S2). Read counts higher than 300 sequences were also observed for the Phylum Mortierellomycota in Chaupi Rodeo B and C locations and in Salinas Grandes soils (Fig. 2), while all other Phyla showed read abundances lower than 100 sequences. Beta-diversity among different locations was assessed by means of Non-metric Multi-Dimensional Scaling (NMDS) ordination system using Jaccard dissimilarity index and statistically tested by means of PERMANOVA (R-squared value of 0.35751 and p-value < 0.001) confirming significant differences among locations, with Salinas Grandes samples clearly being separated from the other locations and the sampling points of Puna and Chaupi Rodeo crops (potatoes, Vicia faba, corn) clustering separately (Fig. 3).

Figure 2 Taxonomic distribution of the retrieved fungal communities at the Phylum level for each soil sample.

Samples are grouped per Location and labelled as follows: Abra = Abra Pampa, CRF = Chaupi Rodeo Vicia faba fields, CRM = Chaupi Rodeo corn fields, CRP = Chaupi rodeo potato fields, Dunas = Dunas, PdM = Puesto del Marqués, PS = Punto Susques, Salar = Salinas Grandes. Phyla are indicated with different colors: green = Ascomycota; orange: Basidiomycota; blue = unidentified fungi; purple = Glomeromycota; light green = Mortierellomycota; yellow = Chytridiomycota; brown = Calcarisporiellomycota; grey = Aphelidiomycota. The letters from A to D identify samples within each location.

Figure 3 Non-metric multi-dimensional scaling (NMDS).

NMDS performed using Jaccard dissimilarity index and statistically tested with PERMANOVA test (R-squared value of 0.35751, p-value < 0.001). Sample locations ellipses are shown, while the labels indicate the macro-environment where each sample was taken.

Occurrence of phylum glomeromycota and ecological drivers

Kruskal-Wallis non-parametric test of variance showed an overall significant difference in AMF abundance among locations (chi-squared = 54.393, p-value = 1.966e–09; Fig. S1) and single samples (chi-squared = 106.21, p-value = 2.411e–11). As a matter of fact, a very scarce occurrence of AMF was observed in the surrounding areas of the salar (Dunas), where we found Glomeromycota sequences only in one sample, while no sequences were found in Salinas Grandes soil samples nor in two of the three potato fields in Chaupi Rodeo (CRP_B and CRP_C). Significant differences in AMF abundances were also observed among habitats, i.e., Puna, salar, corn crops, Vicia faba L. crops and potato crops (chi-squared 37.742, p-value 1.266e–07; Fig. 4), as well as among types of land use, i.e., lama or lama and cattle grazed areas, undisturbed or farmed soils (chi-squared = 27.814, p-value = 3.973e–06; Fig. S2A) and among samples with different land uses in the previous year (chi-squared = 20.993, p-value = 0.0003177; Fig. S2B). Pairwise post-hoc tests with Bonferroni correction (Table S3) revealed that the AMF abundance in the Chaupi Rodeo cropland soils was significantly different from the one retrieved in the soils of the salar (Salinas Grandes) and in the less saline surroundings of Salinas Grandes (Dunas). Only one of the three locations in Chaupi Rodeo (CRP_A) was significantly different from the area of Punto Susques, located within the Puna grassland habitat. The above stated significances among locations were due to the samples CRF_A and CRF_B, both V. faba crops in Chaupi Rodeo, which were significantly different from all Dunas and Salinas Grandes samples, from two Chaupi Rodeo potato crops (CRP_B and CRP_C) and from the samples PdM B and PS_C, these last two both belonging to Puna grassland habitats. Significant differences in AMF reads were also observed among cultivated soils and the soils of the Puna grassland (regardless of being grazed or not grazed by lama) and among the soils of the Chaupi-Rodeo V. faba crop fields and all the other soils as well as among corn crops soils and Puna grassland and salar soils.

Figure 4 Barchart of Glomeromycota read means among different habitats or land uses.

Samples habitats and land uses are labelled along the x-axis of the barchart. Different letters indicate pairwise post-hoc test statistical significance differences with Bonferroni correction (p <= 0.05). Note that the Salar column color is not shown because no AMF sequences were retrieved within that habitat. The average number of sequences retrieved (mean AMF reads) is shown on the Y axis.

Among the measured soil biochemical parameters (pH, salinity, percentage of organic matter, soil carbon quantity, available phosphorus), salinity and soil organic matter were the most significant GLMM drivers of AMF occurrence in the sampled areas (Table S4). Salinity showed the biggest size effect and a negative correlation, thus reflecting how an increasing salinity in the investigated soils seemed to limit the AMF occurrence, while increasing the soil organic matter content may benefit AMF growth. The nested random effects of samples within locations showed variance values greater than zero, hence accounting for some spatial autocorrelation of data. Consistent results were obtained by a Multi Model Inference (MMI) approach. The sum of weights of predictors calculated from model weighting and the model-conditional averaged coefficients highlighted how salinity and organic matter appeared to be the most influential drivers of AMF occurrence in the investigated soils, with estimates significantly different from zero (Table 1).

Table 1 Sum of weights of importance of all predictors as calculated from model weighting through Akaike Information Criterion and model-averaged coefficients.

(A) Each value corresponds to the probability that a variable is included in the best predicted model. (B) Conditional averaging resulting from model averaging using the function dredge in the R package MuMIn. Each estimate is tested for statistical significance as different from zero. (asterisk (*): statistically significant).

A	Salinity	Organic matter	Elevation	pH	Assimilable P	
Sum of weights	0.91	0.77	0.55	0.42	0.27	
	
B	Estimate	Std.error	Adjusted SE	Z value	Pr (>|z|)	Signific.	
(intercept)	−8.33502	4.64333	4.64382	1.795	0.0727	•	
Organic matter	1.81588	0.70544	0.70587	2.573	0.0101	*	
Salinity	−27.34102	12.52910	12.53072	2.182	0.0291	*	
Elevation	1.19262	0.73985	0.74026	1.611	0.1072		
pH	−1.16042	1.07924	1.07971	1.075	0.2825		
Assimilable P	0.01587	0.66192	0.66237	0.024	0.9809		
Note:

Significance codes: 0.01*, 0.05•.

Modeling AMF geographical and environmental suitability in Argentinean Puna

A stringent stepwise selection of climatic, edaphic and land use variables was carried out to model the distribution of AMF in the selected areas of Puna and Altoandino. Principal Component Analysis (PCA) additionally helped to select temperature and precipitation variables likely explaining more variance. Four main clusters of sampling points can be observed within the bioclimatic variation space. Precipitation and temperature seasonality seem to influence the cluster including the locations in or around the salar (salar and Dunas), while temperature and precipitation in the cold dry season may be influential in the Chaupi Rodeo area and the temperature in the wettest and warmer period in the Abra Pampa and Puesto del Marqués locations (Fig. 5). The selected bioclimatic variables chosen to run the final set of models were minimum temperature of coldest month (bio6), temperature annual range (bio7), and coefficient variation of precipitation seasonality (bio15), which were added to the soil, vegetation and land use predictors as described above. Among a set of competing models with AUC values larger than 0.975 generated using MaxEnt, a best predictive model was chosen scoring a value of AUC of 0.981 ± 0.005 for the cloglog output and a value of AUC of 0.979 ± 0.008 for the cumulative output. MaxEnt permutation importance of variables (Table S5) revealed that the variables with a larger contribution in explaining the model were temperature annual range (bio7), soil type and land cover, with lower contributions given by elevation, nutrient retention in soils, quantity of nutrients, vegetation type, and precipitation seasonality (bio15). An overall assessment of the cloglog and cumulative outputs can be inferred observing the plots showing the training omission rate and the predicted area as a function of the cumulative threshold (Fig. 6), averaged over the replicate runs, here complementing the AUC value and confirming a viable quality of the models for the predicted area, even more notably when considering the limited number of presence points derived from the sampling design. The cloglog and cumulative habitat suitability maps for the AMF occurrence were generated as an output of the MaxEnt runs, respectively following a probability and a percentage scale. As expected, the cumulative prediction expands the suitability to larger areas than the cloglog model (Fig. 7).

Figure 5 Principal components analysis of the Worldclim bioclimatic variables performed on sampling points.

(A) Worldclim temperature (bio1 to bio11) and precipitation (bio 12 to bio19) bioclimatic variables vectors are shown for the first two components (PC1 and PC2). The colors indicate a gradient of contribution importance per vector in explaining the variance. (B) The ordination of the sampling points is plotted along with the bioclimatic vectors. The labels refer to the different locations as follows: Salar = Salinas Grandes; Dunas = Dunas; Abra = Abra Pampa; PdM = Puesto del Marqués; PS = Punto Susques; CRP = Chaupi Rodeo potato fields; CRF = Chaupi rodeo Vicia faba fields; CRM = Chaupi Rodeo corn fields. The three Chaupi Rodeo farming sites A, B and C are also indicated in the labels.

Figure 6 Training omission rate and predicted area performed in MaxEnt as a function of cumulative threshold.

Thresholds were averaged over the replicate runs respectively for cloglog output (A) and for cumulative output (B).

Figure 7 AMF environmental suitability as developed by MaxEnt iterative approach on the chosen best model.

(A) Cloglog MaxEnt suitability map on a probability scale; (B) Cumulative MaxEnt suitability map on a percent scale. Probabilities of occurrence are better represented by the cloglog output while habitat suitability is better represented by the cumulative output. Colors indicate a gradient from minimum suitability (deep blue) to maximum suitability (red).

A threshold value was chosen among those calculated by MaxEnt for each of the ten model repetitions to plot a potential presence-absence map. Only thresholds corresponding to low or null omission rates and whose binomial tests of omission rates were statistically significant (p-value < 0.05) were considered as possible presence-absence cut-offs. The MaxEnt threshold generated as a balance among training omission, predicted area and threshold value was applied to the cloglog output, calculating the effective cutoff as a mean of all model repetitions threshold values, corresponding to a cutoff value of 0.0644, the areas below which were then considered as AMF non-idoneity zones and plotted in black on the resulting map (Fig. 8A). A mean of all model repetitions values of 10 percent fixed cumulative thresholds transformed into cloglog probability values generated by MaxEnt was chosen as the effective cutoff for the cumulative output, corresponding to a value of 0.1266, the areas below which were also in this case considered as AMF non-idoneity zones and plotted in black on the cumulative output map (Fig. 8B). Actual absence points (resulting from the OTU table analysis) were not used in this process due to a non-systematic coverage of the predicted territory, thus preferring not to introduce biases in the MaxEnt only-presence environmental suitability predictions.

Figure 8 Binarized environmental suitability maps using all the points of presence.

(A) MaxEnt cloglog model (binarization threshold: 0.0644) (B) Cumulative model outputs (binarization threshold: 0.1266). Black areas correspond to non-idoneity zones for AMF. Other colors represent a blue-red gradient from low to high environmental suitability for the occurrence of AMF. Presence points corresponding to the georeferenced sampling points where Glomeromycota sequences were found are plotted on the map as colored dots as follows: brown = farmed soils, yellow = puna, white = salar.

In a further set of models excluding Chaupi Rodeo points of presence, the values of importance of predictors were consistent with the full points model, except for elevation and type of vegetation. In these models, elevation exhibited negligible importance and the type of vegetation was uninformative, because the used points of presence were all included in the Puna region. Temperature annual range (bio7) and soil type appeared to be the best predictors also in this new set of models (Table S6). The output maps generated excluding Chaupi Rodeo points evidenced a lower AMF suitability of the area of Chaupi Rodeo relative to the AMF suitability previously predicted by the full point model, suggesting that the agricultural land use may be a main driver of AMF distribution in this area. The threshold types chosen as presence-absence cutoffs for this last set of models were the same selected for the full points models, respectively with values of 0.0474 and 0.1143 for the cloglog and the cumulative models (Figs. 9A and 9B). According to the response curves of the generated MaxEnt models, the selection of climate predictors indicates a positive effect of temperature seasonality in modeling the distribution of AMF in the Argentinean Puna, while a higher altitude appears to be negatively correlated with the presence of AMF (Fig. 10).

Figure 9 Binarized environmental suitability maps generated using a subset of points of presence that exclude Chaupi Rodeo points.

(A) MaxEnt cloglog model (binarization threshold: 0.0474) (B) Cumulative model (binarization threshold: 0.1143). Black areas correspond to non-idoneity zones for AMF. Other colors represent a blue-red gradient from low to high suitability of occurrence of AMF. Other colors represent a blue-red gradient from low to high environmental suitability for the occurrence of AMF. Presence points corresponding to the georeferenced sampling points where Glomeromycota sequences were found are plotted on the map as colored dots as follows: brown = farmed soils, yellow = puna, white = salar. The points of Chaupi Rodeo are represented here for reference even if not included in MaxEnt models.

Figure 10 MaxEnt generated response curves for variables bio7 (temperature annual range), soil type and elevation extracted from the chosen cloglog full points model.

The curves show the effect of varying the chosen variable on the MaxEnt prediction, by keeping all other environmental variables at their average sample value. Other response curves can be found in Figs. S4–S7.

Discussion

Environmental metabarcoding outcomes indicated a high occurrence of Ascomycota and Basidiomycota in the investigated Argentinean Puna and Altoandino soils, as expected given the wide taxonomic diversity of these Phyla, and of the Phylum Mortierellomycota with lower number of reads. Even if other studies assessed a widespread dominance of Chytrids in high-elevation periglacial soils (Freeman et al., 2009), we found very scarce occurrence of these taxa in Puna soil, likely due to an endemic aridity in most of the year which may not favor the Chytrid zoosporic reproductive stages. The Phylum Glomeromycota occurred at higher abundances in croplands than in uncultivated Puna and salar soils, likely due to a greater density of host plants in the former ones relative to the typical scarce vegetation of the habitats of these areas. It is worth noting that the sampling sites are mostly included in the so-called Dry Puna (Troll, 1959; Troll, 1968), where annual rainfalls range between 100 mm and 400 mm, and are restricted to the summer season with consequent very dry and cold winters. Although Chaupi Rodeo geographical placement overlaps Altoandino boundaries, these investigated sampling sites in Altoandino are not placed at remarkably higher elevations than the other sites in Puna, therefore they present the same climatic conditions of the other sampled areas.

Enhanced mycorrhizal dynamics and nutrients exchange certainly contributed to shaping different physical and biochemical features in Chaupi Rodeo farmed soils, which were not intensively managed and likely turned out to be more fit to Glomeromycota growth than droughty and poorly evolved soils of Puna environments. Still, this is not a univocal trend, since potato crop fields showed a very low occurrence of AMF not only relative to corn and fava beans but also in comparison to other Puna soils, AMF being virtually absent in two potato crop samples out of three. In a previous study focused on the Chaupi Rodeo cropland soils, Ontivero et al. (2020) underlined that AMF communities were significantly shaped by calcium and nitrogen concentration in soils, despite by extending their analysis to Puna grasslands and hypersaline salar basins we did not find evidence of a significant impact of these elements on AMF distribution. Further studies are therefore required. Similarly, other studies highlighted that pH and calcium concentration were strongly correlated with fungal richness in soils, with particular significance of pH and evapotranspiration for Glomeromycota occurrence (Tedersoo et al., 2014). In a research on the realized niche of AMF, Davison et al. (2021) showed that pH and temperature were the most significant drivers of the global AMF distribution, reinforcing the importance of climatic and biogeochemical features of soils for explaining the AMF distribution. Our findings point to the soil salinity and to the soil organic matter content as the main drivers of AMF distribution when considering all the three macro-habitats considered (grasslands, hypersaline basins, family-run cropfields), the first parameter proving detrimental to AMF abundance and the second one likely enhancing AMF occurrence. Accordingly, studies targeting extreme cold and dry environments within Antarctica valleys (Jiang et al., 2022; Arenz & Blanchette, 2011) observed a significant negative correlation between increasing salinity and microbial and fungal diversity and abundance, while higher carbon and nitrogen concentrations seemed to enhance fungal occurrence. Although the microbial and fungal response to salinity might be habitat and taxon dependent, it is likely that rates of growth decline as the challenges in maintaining a trade-off between physiological homeostasis and gathering resources increase (Okie et al., 2015). Another explanatory framework may be that in these harsh environments increasing salinity can have an indirect effect on fungi through a negative impact on primary producers occurrence (Arenz & Blanchette, 2011).

Among all locations within Puna grasslands, Dunas samples and, to follow, both Puesto del Marqués and Punto Susques samples showed the lowest occurrence of AMF. If the higher salinity of Dunas soil might additionally contribute to lower the AMF presence, for all the above mentioned locations the AMF scarcity is also likely to be impacted by the habitat disturbance due to grazing, flocks of llamas and cattle being restricted in paddocks or in corrals in Puesto del Marqués, and only llamas in Dunas and Punto Susques. In our research we didn’t find significant evidence of a direct effect of grazing on the AMF distribution, this likely due to our sampling effort whose focus was a more general comparison among the three different Puna macro-habitats of salar, grassland and cropfields in native communities. However, other researchers studying AMF responses to grazing with a cellular and physiological approach (van der Heyde et al., 2017), highlighted the importance of the extent of grazing time on AMF occurrence. In Puna, anthropogenic activity since the late Holocene, and for more than 2,000 years before the present, led to intense grazing representing a major cause of shrubland expansion to the detriment of grassland habitats. Furthermore, grazing by domestic animals including cows and camelids proved to be a main driver of more drastic effects on vegetation than those caused only by wild camelid species (Carilla, Grau & Cuello, 2018; Quiroga Mendiola & Cladera, 2018), thus impacting on soil fungal dynamics and likely on AMF distribution as well. The effects of grazing on AMF are however controversial because the responses of these biotrophic symbiotic fungi to herbivory are context-dependent and may be directly related to the carbon flux within the plant-AMF-soil network and to the intensity and extent of grazing over time, as well as to the specific mycorrhizal dependence of the given grazed plants and the adaptation of AMF and their host plants to grazing (van der Heyde et al., 2019; Faghihinia et al., 2020; Klironomos, McCune & Moutoglis, 2004; Hart, Reader & Klironomos, 2003). The relationship between AMF and soil total fungal profile with grazing-shaped vegetation dynamics in Puna ecosystems requires, therefore, further investigation.

The occurrence of AMF in hypersaline salar areas and surroundings, notably the locations of Salinas Grandes and Dunas was very scarce. No AMF sequences were recorded in any of the Salinas Grandes soils, while only one sampling point in Dunas displayed the occurrence of a single Diversisporaceae OTU. As the model averaged estimates statistically suggest, it is therefore likely that high levels of salinity in soils can have a negative impact on the overall AMF occurrence and abundance in these environments, this effect extending also to areas in moderately saline surroundings of the salar. It must be noted however that Glomeromycota are obligate biotrophic symbionts of most vascular plants and the observed variability in abundance and distribution of sequences from these fungi among different points of the same locations might be partly driven by the chance that sampling occurred at different distances from nearby plants or plant root residuals and to plant-host specificity at the ecological group level (Öpik et al., 2009). However, results obtained so far in AMF morphological studies in saline environments of Argentina did not show a clear pattern with regard to the effect of plant identity on AMF occurrence and abundance. Different native halophytes of Salinas of central Argentina showed in their rhizosphere a low diverse AMF community, with AMF species inconsistently varying in their sporulation among soil samples, seasons, plant species and soil depth (Becerra et al., 2014; Cofré et al., 2012; Soteras et al., 2012). Therefore, the effect of salinity on AMF occurrence is not straightforward.

Along with salinity, the abundance of organic matter in soil is suggested to represent a significant predictor of AMF distribution in the investigated soils. Accordingly, MaxEnt outputs assigned the highest explanatory importance to soil type, nutrient abundance and nutrient retention variables, which are clearly related to different organic matter contents in different soils. Even if Glomeromycota are not saprotrophs, a larger amount of organic matter in soil may be an indicator of enhanced nutrient soil dynamics generated by a greater density of plants, thus once more likely explaining the abundance of AMF in crop soils compared to soils of Puna and salar, typically poor in organic matter. MaxEnt response curves regarding edaphic and vegetation variables (Figs. 10 and S4–S7) pointed out that grasslands, followed by sparse herbaceous and shrubs areas, are more suitable for the occurrence of AMF than bare areas, as expected for a taxon in obligate symbiosis with plant roots. Soil types predominantly associated with a greater occurrence of AMF in Puna habitats are luvic yermosols, with a weak ochric A horizon and an argillic B horizon, aridic moisture regime and low organic carbon content, this result mainly corresponding to the overall soil characteristics of Puna habitats. In a recent research Větrovský et al. (2019) assessed that climate is the primary environmental factor for the overall distribution of Fungi. Specifically, temperature proved to be a main driver for AMF distribution, as shown by Zhao et al. (2019) and Davison et al. (2021).

In Argentinean Puna sites, MaxEnt response curves also suggested that a wider annual temperature range showed a greater positive effect on the AMF distribution among the other bioclimatic predictors of AMF seasonality, while higher altitude was negatively correlated with AMF presence. The fact that an increasing temperature seasonality can be related to a greater occurrence of AMF might appear counter-intuitive because it suggests more extreme climatic conditions. The effect of temperature on AMF is related directly or indirectly to carbon and phosphorus AMF-host exchange and to translocations from soil to the host plants (Gavito et al., 2005). Recent research highlighted that AMF showed differential patterns of growth depending on the environmental conditions of their habitats, with a higher resistance to high temperature in arid and semiarid ecosystems (Kilpeläinen, Aphalo & Lehto, 2020). Therefore, in the ecological context of the Puna, it can be inferred that in the wet and warm season AMF can benefit from higher temperatures, enhanced water availability and metabolic activity, likely remaining dormant as spores in less favorable periods. The seasonal distribution of AMF is well known for their sporulation (Smith & Read, 2008), measured in terms of spore abundance and sporulating species richness and root colonization, especially for other investigated South American highland grasslands such as the Pampa de Achala (Lugo & Cabello, 2002; Lugo, González Maza & Cabello, 2003) and different mountain environments (Soteras et al., 2019). The altitude negative trend might be related to less favorable environmental and climatic conditions at higher altitudes for plants and therefore for AMF, as it has been reported in Argentinean Puna considering sporulation (Lugo et al., 2008) and root colonization (Lugo et al., 2012). It must be reminded that according to our experimental design, the Chaupi Rodeo sampling points fall within the Altoandino and despite they weren’t collected at higher elevations than the other sampling points, the edaphic profile of that area is different, showing a prevalence of lithosols instead of the above described yermosols. A study performed in the Andean Yungas Forest ecoregion (Geml et al., 2014) assessed how soil-fungal community structure is negatively correlated with elevation, in accordance with former studies on AMF distribution on Himalaya (Gai et al., 2012) and our results on AMF sequences distribution. Although a wider sampling in high elevation gramineous grasslands might be required to better explain the AMF ecology in these habitats, AMF community richness have already been assessed to be higher in grasslands at global (Tedersoo et al., 2014; Davison et al., 2015) and local scales (Grilli, Marro & Risio Allione, 2019). We also acknowledge that further studies focused on AMF reads, i.e., using specific primers, might help testing whether future results in the macro-habitats of Argentinean Puna will be consistent with the relative occurrence of AMF within the soil fungal profile that we assessed in our research.

Given the evidence that croplands hosted a much greater abundance of Glomeromycota, we considered it advisable to account for the human intervention in cultivated soils in the modeling process of AMF distribution. Setting and selecting a new series of models excluding the cultivated Chaupi Rodeo samples allowed us to model the potential AMF distribution as if these areas had virtually not been cultivated, that is verifying how large-scale soil features and bioclimatic constraints would act on the AMF distribution if these soils had remained unmanaged habitats of Puna and Altoandino. Such a new final model unveiled a much lower suitability of Chaupi Rodeo location for Glomeromycota than previously modeled. At a wider scale it also reshaped the distribution of AMF in the areas where climatic and soil type conditions were similar to Chaupi Rodeo locations. It is noteworthy that the new models excluded lithosols from high suitability: as said, these soils are typically found in the unfarmed Chaupi Rodeo areas and in the previous models they were inversely considered as good predictors of AMF occurrence (see Figs. S4–S6). This is easily explained considering that even if lithosols are dominant in the Chaupi Rodeo area, the agricultural settlements have progressively altered the soil profile at least within the crop fields. This confirms that shaping distribution models without accounting for human activity in the studied landscape may prove misleading but likewise this allows to make comparisons with models which include the human impact, therefore helping to predict habitat suitabilities more accurately and in a wider ecological perspective.

Conclusions

To sum up, we performed an exploratory assessment of AMF environmental suitability in an ecologically homogeneous area of Argentinean Puna and Altoandino throughout three different representative macro-ecosystems: Puna grasslands, a hypersaline endorheic basin and family-farmed croplands. We concluded that the differential human impact on frail ecosystems like the Argentinean Puna should be assessed also for soil and microbiological dynamics, especially when investigating deep plant-related organisms like AMF so essential in preserving not only the ecosystems stability but also the farming productivity, this last unreplaceable in the subsistence-based human communities of Puna. Likewise, AMF distribution may be a good indicator of the changes that human intervention triggers in the Puna ecosystems, such as crop cultivation. We therefore suggest that in future research they may also be useful for investigating alterations of the natural ecosystem equilibrium in those locations of Puna where exceeding intensive grazing, logging and mining activities threaten these unique ecosystem’s functioning. We encourage further investigations of the structure and composition of the soil mycobiota of Puna at a larger scale and with an in-depth assessment of the differential impact of human activities on the many habitats composing this nowadays endangered biogeographical region.

Supplemental Information

Supplemental Information 1 Barplots of Glomeromycota read means among locations.

The sum of sample reads is grouped and plotted per location. Samples locations are indicated by colors as described in the legend and labelled below each bar. Different letters indicate pairwise post-hoc test statistical significances with Bonferroni correction (p <= 0.05). Note that the Salinas Grandes column color is not shown because no AMF sequences were retrieved within that location. The average number of sequences retrieved (mean AMF reads) is shown on the Y axis.

Click here for additional data file.

Supplemental Information 2 Barplots of Glomeromycota read means among different land uses (A) and among sample groups with different land uses in the previous years (B).

Colors refer to the contribution of the different tested groups. Group names are labelled below each bar. Different letters indicate pairwise post-hoc test statistical significance difference with Bonferroni correction (p <= 0.05). Note that some column colors are not shown because no AMF sequences were retrieved within these groups. The average number of sequences retrieved (mean AMF reads) is shown on the Y axis.

Click here for additional data file.

Supplemental Information 3 MaxEnt full-point cloglog response curve (mean output) for the variable land cover.

The curves show the effect of varying the chosen variable on the MaxEnt prediction, by keeping all other environmental variables at their average sample value. Labels of the highest response values are indicated.

Click here for additional data file.

Supplemental Information 4 MaxEnt full-point cloglog response curves (mean output) for the chosen variables: bio7, bio15, elevation, land cover, nutrient retention, nutrient quantity, soil type and vegetation type.

The curves show the effect of varying the chosen variable on the MaxEnt prediction, by keeping all other environmental variables at their average sample value. Worldclim bioclimatic variables Bio7 and Bio15 represent respectively the temperature annual range and the precipitation seasonality.

Click here for additional data file.

Supplemental Information 5 MaxEnt full-point cumulative response curves (mean output) for the chosen variables: bio7, bio15, elevation, land cover, nutrient retention, nutrient quantity, soil type and vegetation type.

The curves show the effect of varying the chosen variable on the MaxEnt prediction, by keeping all other environmental variables at their average sample value. Worldclim bioclimatic variables Bio7 and Bio15 represent respectively the temperature annual range and the precipitation seasonality.

Click here for additional data file.

Supplemental Information 6 MaxEnt reduced-point cloglog response curves (mean output) for the chosen variables: bio7, bio15, land cover, nutrient retention, nutrient quantity, soil type.

The curves show the effect of varying the chosen variable on the MaxEnt prediction, by keeping all other environmental variables at their average sample value. Worldclim bioclimatic variables Bio7 and Bio15 represent respectively the temperature annual range and the precipitation seasonality.

Click here for additional data file.

Supplemental Information 7 MaxEnt generated reduced-point cumulative response curves (mean output) for the chosen variables: bio7, bio15, land cover, nutrient retention, nutrient quantity, soil type.

The curves show the effect of varying the chosen variable on the MaxEnt prediction, by keeping all other environmental variables at their average sample value. Worldclim bioclimatic variables Bio7 and Bio15 represent respectively the temperature annual range and the precipitation seasonality.

Click here for additional data file.

Supplemental Information 8 Total number of sequences summarized per location and Phylum.

Count = number of OTUs; Sum = sum of reads.

Click here for additional data file.

Supplemental Information 9 Number of Glomeromycota sequences summarized per sample and location.

Count = number of OTUs; Sum = sum of reads.

Click here for additional data file.

Supplemental Information 10 Pairwise post-hoc tests with Bonferroni correction.

Tests were performed on Glomeromycota reads among locations of the sampled areas (A) and among the levels of the following factors: soil disturbance (B), habitat/crop type (C), preceding crop/habitat in the previous year (D). Below (E) are shown the statistical significance of differences among samples.

Click here for additional data file.

Supplemental Information 11 Generalyzed Linear mixed model output.

(A) Generalized Linear Mixed Model estimates of fixed effects generated through a Minimum Adequate Model (MAM) approach. Significance codes: *** ‘0’, ** ‘0.001’, * ‘0.01’, ⋅ ‘0.05’. (B) Spatial autocorrelation of data was accounted for by the nested random effect of samples within locations, showing variance greater than zero. Number of observations: 1,736. Groups: Sample:Location, 28; Location, 8.

Click here for additional data file.

Supplemental Information 12 Values of importance of variables calculated in the MaxEnt models with all the presence points.

Variables importance was calculated by MaxEnt as a percent contribution and as a percent contribution after permutation of all variable values. (A) Cloglog output table (B) Cumulative output table. The considered variables were bio7, bio15, land cover, soil, elevation, nutrient retention, quantity of nutrients, vegetation type. AUC scores are 0.978 sd 0.005 for the cloglog model and 0.978 sd 0.007 for the cumulative model.

Click here for additional data file.

Supplemental Information 13 Values of importance of variables in the MaxEnt models devoid of Chaupi Rodeo sampling points.

Variables importance was calculated by MaxEnt as a percent contribution and as a percent contribution after permutation of all variable values. (A) cloglog output table; (B) cumulative output table. The considered variables were bio7, bio15, land cover, soil type, nutrient retention, quantity of nutrients. AUC scores are 0.981 sd 0.003 for the cloglog model and 0.983 sd 0.003 for the cumulative model.

Click here for additional data file.

Supplemental Information 14 Metadata of soil sampling for BioProject PRJNA835719.

Click here for additional data file.

Supplemental Information 15 Feature table (19882 sequences per sample) raw data.

Click here for additional data file.

Supplemental Information 16 Taxonomy raw data.

Click here for additional data file.

Additional Information and Declarations

Competing Interests

Author Contributions

Field Study Permissions

Data Availability

Erica Lumini is an Academic Editor for PeerJ.

Davide Nepote Valentin analyzed the data, prepared figures and/or tables, authored or reviewed drafts of the article, and approved the final draft.

Samuele Voyron analyzed the data, prepared figures and/or tables, authored or reviewed drafts of the article, and approved the final draft.

Florencia Soteras analyzed the data, prepared figures and/or tables, authored or reviewed drafts of the article, and approved the final draft.

Hebe Jorgelina Iriarte analyzed the data, prepared figures and/or tables, authored or reviewed drafts of the article, and approved the final draft.

Andrea Giovannini analyzed the data, authored or reviewed drafts of the article, and approved the final draft.

Erica Lumini conceived and designed the experiments, performed the experiments, analyzed the data, authored or reviewed drafts of the article, and approved the final draft.

Mónica A. Lugo conceived and designed the experiments, performed the experiments, analyzed the data, authored or reviewed drafts of the article, and approved the final draft.

The following information was supplied relating to field study approvals (i.e., approving body and any reference numbers):

Gobierno de Jujuy, Ministerio de Ambiente, Jujuy Argentina.

The following information was supplied regarding data availability:

The ITS2 raw sequences data are available at the NCBI Sequence Read Archive: PRJNA835719.

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
