# Peer review of "Modeling geographic distribution of arbuscular mycorrhizal fungi from molecular evidence in soils of Argentinean Puna using a maximum entropy approach"

_PeerJ, doi:10.7717/peerj.14651_

## Round 0.1 · original submission · Major Revisions

Dear Authors,
We have received initially a <Minor Revisions> recommendation from one of the reviewers, followed by a <Reject> recommendation from the subsequent reviewer, Then, I asked two more Reviewers in order to give a decision as Associated Editor. These two other reviewers showed distinct kinds of recommendations: one <Minor Revisions> and the other <Major Decisions>. Therefore, I recommend this submitted article as <Major Decisions>. The questions raised by the Reviewer that rejected the paper must be especially thoroughly replied to, as well as all the other questions from the other reviewers.

Reviewer 1 has suggested that you cite specific references. You are welcome to add it/them if you believe they are relevant. However, you are not required to include these citations, and if you do not include them, this will not influence my decision.

Reviewer 1 ·

Basic reporting

In general, the article is well structured and done, and accounts for a great research effort, with important international collaboration.
Moderate English changes required, I believe English is their second language and as such the paper will need some editing to bring out the brilliance of the research.
Some literature references reviewing is needed to complement ideas and support some results, this is indicated in annotated PDF.
Figures need more detail and better resolution, specially in legends.
Several parts of the article seems to be beyond the main research question, is important to clarify your research question and describe your findings around it.

Experimental design

Article compiles meets the objectives and scope of the journal.
The authors need a review to clarify the research question and minimize or eliminate information beyond their research. Clarify, what is the objective for modeling geographic distribution of AMF?
The Authors present a rigorous and high technical investigation process and cooperative research effort.
Authors need to clarify AMF modeling parametrization: when software run with defaults paremeters, etc. ? This to allow replication of their research method

Validity of the findings

The results of this research strengthen the information on species modeling from unusual perspectives, such as the molecular and multiscale perspective. This will allow this type of study to be better and more frequent.
Some changes in the conclusions are suggested, which will definitely strengthen the excellent results of this research. Suggested changes in the introduction, and their use in discussions may help.

Additional comments

I have made comments throughout the paper on the pdf. Please review these comments. Some of the comments refer to phrases that can be removed to clarify a sentence. Some comments address sentence structure that need to be improved and content of the sentence improved.

Annotated reviews are not available for download in order to protect the identity of reviewers who chose to remain anonymous.

Reviewer 2 ·

Basic reporting

The manuscript describes AM fungal communities in Puna ecosystems in South America, but with no specific hypothesis and research questions

The English language use is mostly understandable, though clearer phrasing could be used and grammatical inconsistencies exist (e.g,, “plants growth” vs plant growth, L 119; “sampling sites can be included in Puna” vs sampling sites were located in Puna, L159; etc)

The text is poorly structured. For example, the introduction is a single paragraph. An easy to read text would present every point as a separate paragraph

Experimental design

Introduction spends most of space on detailed description of the study system. This is mostly an interesting read, but it does not formulate a research questions and hypotheses, nor does it explain why conducting this study was meaningful. The statement that “there is a paucity of research addressing fungal profile in Puna soils” (L140) is not sufficient to explain why fungal communities should be explored therein. Lack of clearly defined research question is a large limitation of this study
Next, the study design is hard to follow. Firstly, which data are previously published (L143) and which are new? How many study sites there were, 8 or 6 (L165-169 I count 8 location names, but on maps there are 6 points)? How was the soil sampled for AM fungal assessment and for biogeochemical analyses?

Further, distribution of study sites among ecosystem types seems to be very uneven: if exploring ecosystem type effect on AMF is one of the aims, then each ecosystem type should be represented by multiple locations. Again, number of locations included in the data analysis is not clear. In the results section L310-317 suggest that more than 6 sites were studied. It is not possible to assess the results if no clear information is provided about the source data structure.

A serious limitation is that soil samples used for AMF identification were very small: 250 mg. note that currently many soil fungal studies use 2 to 5 g of soil, or sometimes even 10 g.

It is also not clear which primers were used by the authors: ITS region primers, but please provide primer names. If the protocol of Berruti et al (2017) was used , then they chose general fungal primers designed by Ihrmark et al (2012). As the results section shows, majority of reads were non-AMF reads; hence, the less abundant AMF in extreme habitats might have gone undetected due to the primer choice. In other words, the conclusions that in some of the sampled locations AMF were not existing, or that these habitats were not suitable for AMF, might be caused by suboptimal assessment of AMF presence and diversity, rather than based on fair understanding of AMF occurrence. Lekberg et al have assessed the very same primer choice as Berruti et al made (comparison of general fungal ITS primers vs AMF-specific SSU primers), concluding that more detailed data on AMF diversity can be gained only with AMF specific primers.

Thus, my conclusion is that (1) is not currently possible to assess the worth of the data due to missing information about the data structure (number of data points, their distribution), and (2) the conclusions might be strongly influenced by suboptimal analysis of AMF data from general fungal diversity data in the ITS dataset

Validity of the findings

see above

Additional comments

none

Reviewer 3 ·

Basic reporting

General comments:
This paper focuses on investigating the AMF distribution and diversity changes in extreme environments (i.e., aridity and low temperatures) of Argentinean Puna using Illumina amplicon sequencing and maximum entropy approaches.
The research is original, and the research questions are well defined. The methods in this research are novel and sufficient to answer the research questions.
Based on the current manuscript quality, this manuscript should be accepted with minor revisions.

Specific Comments:
Abstract:
Authors only need to cover the most important findings in their abstract. The number of words in the abstract seems to exceed the journal’s requirement. Consider writing it concisely and meeting the Peer J requirement.
Introduction:
Generally, the goal of the study is clarified. There is sufficient background information about this research and the background information is well connected to the current study.
Currently, the whole “Introduction” has only one paragraph. I suggest that authors break down one “Introduction” paragraph into 3 or 4 paragraphs. This will help readers to understand easily. Particularly, authors should list their current research goals/questions/hypotheses at the end of the Introduction section

Materials and Methods:
Generally, more information is needed.
Line 164: It is not clear how these 28 samples from different sampling sites (extreme environments) were preserved after collection. I suggest authors include a brief description with some citations. Here are some papers that can be cited (samples also collected from extreme environments).
Jiang, X., and Takacs-Vesbach, C.D. (2017) Microbial community analysis of pH 4 thermal springs in Yellowstone National Park. Extremophiles 21: 135-152.
Van Horn, D.J., Wolf, C.R., Colman, D.R., Jiang, X., Kohler, T.J., McKnight, D.M. et al. (2016) Patterns of bacterial biodiversity in the glacial meltwater streams of the McMurdo Dry Valleys, Antarctica. FEMS Microbiol Ecol 92.
Jiang, X., et al. (2022). "Limits to the three domains of life: lessons from community assembly along an Antarctic salinity gradient." Extremophiles 26(1): 1-14.

Line 182: “the ITS2 region”. The primer names and sequences to amplify the ITS2 region should be given here.
Line 186: “BMR Genomics”. If “BMR” is the abbreviation of a company name, consider giving the full name of “BMR Genomics”

Discussion:
Some suggestions for the discussion:
The current discussion is forced on what they have found (AMF distribution) in the soils of Argentinean Puna. However, the authors should extend their discussion to other similar extreme environments on other continents. For example, the current study sites are acidic, high UV, high salinity, and cold sites. McMurdo dry valleys, Antarctica, has similar but more extreme environments (high salinity, cold and high UV). Authors can also extend their discussion to the other microbiome (e.g., Bacteria, Archaea, etc.) either in Argentinean Puna or other alpine or similar extreme environments in Europe, Asia, or Antarctica. I hope the authors can add one paragraph in “Discussion” to do a broader discussion/comparison. This will make the discussion more comprehensive. I list some published papers for authors to cite and refer to.
Van Horn, D.J., Wolf, C.R., Colman, D.R., Jiang, X., Kohler, T.J., McKnight, D.M. et al. (2016) Patterns of bacterial biodiversity in the glacial meltwater streams of the McMurdo Dry Valleys, Antarctica. FEMS Microbiol Ecol 92.
Jiang, X., et al. (2022). "Limits to the three domains of life: lessons from community assembly along an Antarctic salinity gradient." Extremophiles 26(1): 1-14.

Hotaling, S., et al. (2019). "Microbial assemblages reflect environmental heterogeneity in alpine streams." Global Change Biology 25(8): 2576-2590.
Mingjia Li, Jinfu Liu, Jonathan D. Tonkin, Ji Shen, Nengwen Xiao, Jianjun Wang, The effects of abiotic and biotic factors on taxonomic and phylogenetic diversity of stream epilithic bacteria around Qiandao Lake, Aquatic Sciences, 10.1007/s00027-020-00746-8, 82, 4, (2020).

Tables and Figures:
Figure 1 and Figure 2: The size of fonts on some maps are too small to see. The size of fonts is not consistent on Figure1 and Figure 2. Consider revising.

Experimental design

Please see the Basic reporting

Validity of the findings

Please see the Basic reporting

Annotated reviews are not available for download in order to protect the identity of reviewers who chose to remain anonymous.

Reviewer 4 ·

Basic reporting

This study investigated mycorrhizal fungi from soil samples collected from various spots in Argentinean Puna. Data was analyzed extensively. English language is OK in this manuscript. Small typos were included in my specific comments. Although ‘Modeling’ is included in the title, I recommend authors explain more results of this study based on the figures/tables acquired from the models/analysis.

Experimental design

How many replications of soil samples were collected in each of the 28 sampling points?

Validity of the findings

Some supplementary figures should be included in the manuscript. However, there are too many figures in total. I would recommend authors merge similar or relevant figures in one big figure and make some selections on the figures to be published.

Additional comments

General comments
1. Introduction can be separated into several paragraphs.
2. The sampling site description will be easier to understand coming with a map. However, in Figure 1, abbreviations were not explained very well in legends, making the map difficult to understand. Why ‘Salinas Grandes’ and ‘Dunas’ were shorted as ‘Unto’. Not only CR had 3 points labeled by ABC, but other locations with ABC were also not explained in legend. Besides, in Figure 1, some locations were labelled as abbreviations, some locations were labelled as full name. Most importantly, I cannot find the location/boundary of Puna in the map.
3. Materials and methods included too many redundant descriptions. I understand authors tried to explain what procedures were done on the model building, selection, analyzing, and prediction. However, too much detail made the methods difficult to read for audiences. Especially when comparing descriptions of soil physicochemical analyses methods and statistical analyses, one is simplified, the other is too redundant.
4. In figure 2 legend, genus names and species names needed to be italic. It seems Figure 2 was not explained either in Materials and Methods, or in results. Only based on the legend of figure 2, I can hardly tell what information authors want to provide through figure 2. For example, availability of which nutrient is shown in Figure 2D? Please add citations to the legend of figure 2.
5. In figure 3, on the horizontal axis, why the order of labels within each region is not alphabetical? Are there any special meanings? Please explain the abbreviations of labels on horizontal axis in legends of figure 3.
6. Figure 4: Different levels/meanings of ‘_A’, ‘_B’, and ‘_C’ between Chaupi Rodeo and other locations are very confusing.
7. Figure 5: The purple color of ‘Salar’ cannot be shown in the figure. Similar problems were found in Figure S1, S2B. Please add horizontal axis labels instead of legend on the right of the figure. ‘N-seq’ is equal to ‘Glomeromycota read’? Please increase font size of vertical labels.
8. Since the methods to generalize Figure S1-S6 were mentioned in Materials and Methods, results of these figures were described in the manuscript. I don’t know why authors made them as supplemental files instead of formal figures included in this manuscript.

Specific comments
Line 98: Please superscript ‘2’
Line 143: Please explain ‘OTU’ the first time it appeared in the manuscript.
Line 169-171: Please briefly describe those environmental predictors in each chosen sampling point. This information can be included either in Materials and Methods, or Results.
Line 207: ‘GLMM’ is not proper to appear in the heading.
Line 236-239: ‘Despite’, ‘therefore’s are not proper. Please re-write lines 236-241.
Line 266: Sorry I don’t understand ‘setting the test data to 30%’.
Line 306: Please delete one of the dots.
Line 338-351: Instead of introducing more details of how PCA was performed, audience would expect more ‘results’. For example, in Figure S3A, 3 clusters can be found, what characteristics do they have based on bio 1-10 and elevation? Besides, how much variance can be represented by PC1 and PC2 in Figure S3A and S3B, respectively?
Lines 412, 413 and 416: ‘how’ should be deleted. Please check similar problems in the whole manuscript.

---

## Round 0.2 · Minor Revisions

One of the reviewers still pointed out problems that must be solved. Please, go through them and submit your revision as soon as possible.

Reviewer 3 ·

Basic reporting

Dear Editor,
The manuscript has been significantly improved after the revision. The authors have addressed all my questions and concerns. Therefore, I recommend that this manuscript should be formally accepted and published.
I have one suggestion about this manuscript, and I hope the authors can revise these before this manuscript formally published.
The authors need to go over the format of all cited literature. I notice the format is not consistent. For example, the following citations do not use the same format.
Line 701: Trends in Ecology & Evolution 18(8)
Line 708: FEMS Microbiology Ecology 82(3)

Experimental design

NA

Validity of the findings

NA

Additional comments

NA

Annotated reviews are not available for download in order to protect the identity of reviewers who chose to remain anonymous.

Reviewer 4 ·

Basic reporting

I appreciate the careful revisions made by authors. I accept most of the reponses to my comments. However, there were still 2 small problems listed in additional comments.

Experimental design

no comment

Validity of the findings

no comment

Additional comments

1. The link added to the legend of figure 1 is not accessible. It shows ‘400. That’s an error. Your client has issued a malformed or illegal request. That’s all we know.’
2. Please explain ‘salar’ and ‘punean’, if it’s possible, please use English instead Spanish for similar phrases.

---

## Round 0.3 · accepted · Accept

The authors have fully addressed the reviewer's comments, and I am completely satisfied with the current status of the manuscript, and, thus, I consider that the manuscript is ready for publication. Congratulations!